# Research on the evaluation and impact trends of China's skill talent ecosystem in the digital era – An analysis based on neural network models and PVAR models

Gaoyang Liang[1]*, Minqiang Xing[2]

1 School of Public Administration, Hebei University of Economics and Business, Shijiazhuang, Hebei, China,
2 School of Management, Xi'an Jiaotong University, Xi'an, Shanxi, China

* lianggaoyang@hueb.edu.cn

## Abstract

This study develops a "Skill Talent Ecological Evaluation Model" across cultivation, potential energy, kinetic energy, innovation, and service and support ecologies. AHP-entropy determines indicator weights, Hopfield neural network assesses talent ecology levels, and the PVAR model analyzes digital transformation effects. Findings reveal: Cultivation ecology rates A, potential ecology rates B+, kinetic ecology rates B-, service and support ecology rates B-, and innovation ecology rates C. Digital transformation spurs skill demand, impacting talent and economic contributions. Kinetic ecology sees increased demand, potentially impacting traditional industries positively. Innovation ecology necessitates continuous skill learning. Service and support ecology witnesses growth in digital entrepreneurship, requiring policy incentives and incubation center support.

## 1 Introduction

In the digital era, the necessity and importance of exploring the talent ecosystem have become increasingly prominent, directly impacting the global economic digitization and the driving force behind technological innovation. Firstly, there is a sharp increase in demand for digital skills, making it a core element for countries to gain a competitive advantage in the global digital economy. Secondly, industries driven by data are on the rise, requiring talents with cutting-edge skills in data science, artificial intelligence, and other emerging fields to address continuously emerging challenges. Thirdly, the growing scarcity in the global digital labor market highlights the urgent need for talents with advanced skills in the digital era. Lastly, the construction of the talent ecosystem for digital skills is crucial for a country's position in the global digital economy, as a workforce adaptable to digital work environments will be more competitive. Although precise figures depend on the latest international and national data, the above points emphasize the critical importance of the talent ecosystem for skills in the digital era for the global economic and social development.

**Data Availability Statement:** All relevant data are within the paper and its Supporting Information files. The data on the skilled talent ecosystem and

digital transformation in this paper can be found in the data file "Skilled talent ecosystem and digital transformation.xls".

**Funding:** This paper was supported by the National Natural Science Foundation of China [No. 71872142].

**Competing interests:** No potential conflict of interest was reported by the authors.

Although research in the academic community on the talent ecosystem has made some progress, there are still several **deficiencies and controversies**. Firstly, the definition of the talent ecosystem is vague, with differences in scope and connotation among various researchers, leading to inconsistencies and comparability issues in research results [1]. Secondly, data availability and quality pose bottlenecks in research, as obtaining accurate information on the quantity and structure of global or national talent ecosystems may be constrained [2, 3]. Methodological issues also generate controversy, with some scholars emphasizing quantitative analysis while others lean towards qualitative research, resulting in diverse interpretations of research findings. Timeliness is another concern, as rapid changes in technology and the economy may render research results outdated [4, 5]. Additionally, regional disparities, policy influences, and uncertainties regarding future trends contribute complexity and controversy to talent ecosystem research [6, 7]. In summary, addressing these deficiencies and controversies requires interdisciplinary collaboration to enhance the depth and breadth of research.

In conclusion, this article presents **three potential points of innovation**. First, it introduces the concept of the talent ecosystem innovatively, emphasizing the interdependence, mutual influence, and driving role of skills talents in the industrial ecosystem. Second, it establishes an evaluation index system for the talent ecosystem. Building upon existing evaluations of skills talents, it incorporates five dimensions: nurturing ecosystem, potential ecosystem, kinetic ecosystem, innovation ecosystem, and service and support ecosystem, forming a comprehensive evaluation index system. Third, the article conducts empirical evaluations of the talent ecosystem using entropy method, Analytic Hierarchy Process (AHP), fuzzy comprehensive evaluation, and Discrete Hopfield Neural Network analysis. By combining real data on skills talents and industry with expert scoring data, the article evaluates and analyzes the talent ecosystem, determines the weights of various talent ecosystem factors, and conducts classification and ranking evaluations. Finally, based on the empirical evaluation results, corresponding strategic recommendations are proposed.

## 2 Literature review

"Talent ecosystem" is a novel concept inspired by ecology, possessing the capability to accurately articulate the overall survival state of talents [8]. It primarily focuses on the interaction between talents and the internal and external environments. The ecosystem of skilled talents refers to a research focus on skilled talents, considering them as an organic whole that interacts and depends on natural, social, and economic environments [9, 10]. This ecosystem encompasses internal composition, external environments, characteristics, and functions. It exhibits features of openness, dynamism, and adaptability, with a certain degree of self-regulation. Its core objective is to promote the optimal development of the ecosystem of skilled talents. Academic research on the ecosystem of skilled talents encompasses the following aspects:

### 2.1 The industry ecosystem of skilled talents

Skilled talents play an indispensable and crucial role in industrial clusters, especially against the backdrop of rapid development in the manufacturing sector. The increasing demand for technical skills talents becomes a vital driving force for further advancement in the manufacturing industry [11]. With the rise of collaborative efforts between industry and education and the emergence of cooperative platforms, training institutions and schools are adjusting their specialties to promote collaborative innovation in applied technology [12]. This, in turn, provides talent and technical support to meet the needs of the industry. However, the issue of a shortage of skilled talents is a subject of considerable concern. Research has identified shortages in the skilled talent market, attributed to improper societal value assessments,

flaws in training mechanisms, inadequate training funds, and deficiencies in evaluation and incentive mechanisms [13]. Additionally, some studies explore issues such as the imbalance in the supply and demand of skilled talents, the distribution of skilled talents in urban areas, and the aging trend in the age structure of skilled talents, from both a city-level and an industry structure upgrade perspective.

## 2.2 The cultivation ecosystem of skilled talents

Academic research has focused on different models of "school-enterprise" collaboration for training high-skilled talents [14]. Some scholars, approaching from the perspectives of sociology and management, have investigated the influence of paternalistic leadership on the willingness of skilled talents to share knowledge [15, 16]. In the digital era, there is a growing emphasis on the study of the cultivation ecosystem of skilled talents, with interdisciplinary research shedding light on the societal, cultural, and policy factors influencing skilled talents [17]. Governments, industries, and enterprises are actively involved in constructing the cultivation ecosystem, with policymakers proposing incentive programs, and collaboration between businesses and educational institutions providing practical skills training. Research underscores the importance of practical experience and application, as well as the critical role of assessment and quality control, ensuring the development of skilled talents with enhanced market competitiveness.

## 2.3 The policy ecosystem for skilled talents

Skilled talent policies encompass multiple domains, including education, the labor market, and industrial policies [18, 19]. These policies are diverse, intricate, and interwoven, reflecting the government's comprehensive considerations in nurturing and attracting skilled talents. With the development of globalization, there is an increasing trend in international comparative research on skilled talent policies [20]. Researchers focus on policy practices in different countries and regions to understand the experiences and lessons learned in skilled talent cultivation and mobility worldwide. The digital era necessitates policies with flexibility and innovativeness to adapt to the rapidly changing technological and economic environment. Consequently, there is a growing body of research on how policies can be adjusted to support the adaptability and innovativeness of skilled talents.

## 2.4 Digitalization and the skilled talent ecosystem

The digital transformation has profound implications for the skilled talent ecosystem. Firstly, the digital era rapidly shapes and alters the pattern of skill demand as the continuous emergence of new technologies renders skills in specific fields gradually obsolete while giving rise to entirely new skill requirements [21]. This prompts researchers to focus on the evolutionary process of skill demand. However, the evolution of skill demand also gives rise to skill gaps and inequality issues, leading individuals lacking digital skills to face exclusion, while those possessing such skills find it easier to secure high-paying jobs. Secondly, digital technologies lead to issues of skill matching and mismatching; insufficient or excessive skills may result in inequalities in employment opportunities and wages, significantly impacting both the labor market and individual outcomes [22]. Finally, digital technologies also exacerbate labor market inequalities, widening the skills gap and intensifying labor market segmentation, with digital automation playing a substitutive role in the workforce [23]. These challenges prompt researchers to explore innovative solutions in education, training, skill matching, and social policies to address the challenges brought about by the digital era.

In conclusion, in the era of digital transformation, research on the skilled talent ecosystem is a crucial area. The research landscape needs further exploration and addressing of several key gaps. Firstly, there is a deficiency in the evaluation of the skilled talent ecosystem. Existing studies predominantly focus on the ecosystem evaluation of managerial, technological, and research-oriented talents, with relatively less attention to the evaluation of skilled talents in the manufacturing industry. Given the critical role that skilled talents play across various industries, their cultivation and mobility are paramount for sustainable economic development. Therefore, there is a need to increase research on the evaluation of the skilled talent ecosystem. Secondly, there is a lack of research on the indicator system for evaluating the skilled talent ecosystem. Comprehensive and specific evaluation indicators need to be established for the ecosystem evaluation of skilled talents. This indicator system should consider various factors such as national policies, industry demands, skill training, and incentives. Thirdly, there is insufficient quantitative research on the evaluation of the skilled talent ecosystem. Most existing studies lean towards qualitative theoretical analyses, with relatively fewer focusing on quantitative research for the evaluation of the skilled talent ecosystem. Quantitative research can more accurately measure the impacts of different processes and mechanisms, as well as assess the sensitivity and degree of change under different policies and economic variations. In summary, the research field of skilled talent ecosystem evaluation requires more attention and in-depth investigation to ensure the growth and development of skilled talents align with current societal and economic needs. This will contribute to enhancing national competitiveness, promoting sustainable development, and improving the professional prospects and social status of skilled talents.

## 3 Theoretical foundation and model construction

### 3.1 Theoretical model for skilled talent evaluation

The essence of evaluating skilled talents lies in determining their ecological position within the talent ecosystem of a specific professional domain and employing specific methods to identify, assess, and manifest the ecological capacities and dynamics represented by this position (See **Fig 1**, for

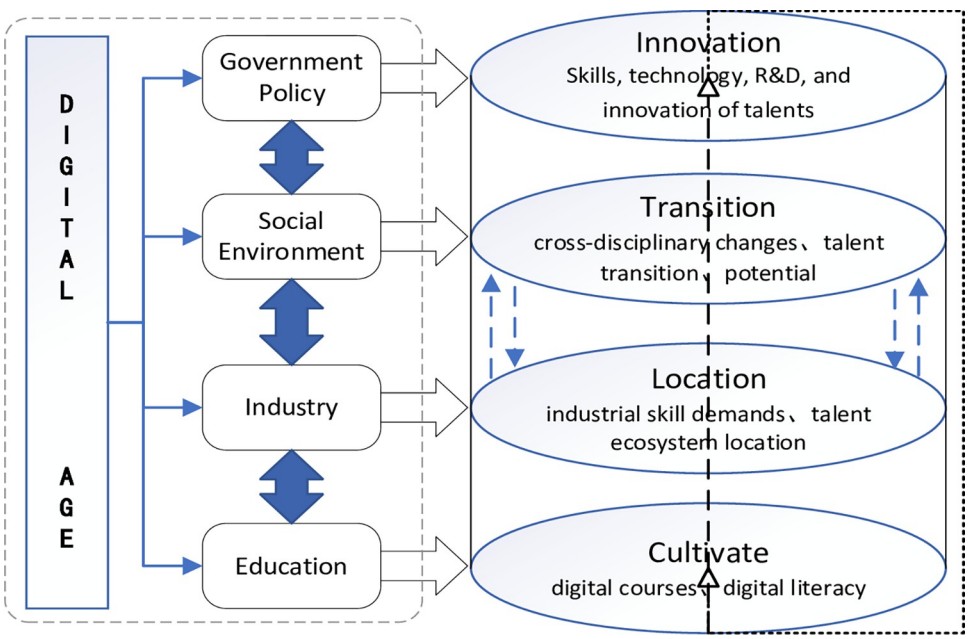

**Fig 1. The impact of the digital era on the skilled talent ecosystem.**

details). *(a)* The potential of the skilled talent ecosystem primarily refers to the specific value, capabilities, contributions, influence, skill levels, and impact of talents within their domain. *(b)* The dynamics of the skilled talent ecosystem mainly pertain to the oscillation and upward potential of talents in their ecological positions, representing their developmental and advancement potential.

*(c)* The cultivation ecosystem refers to an organic system established for the nurturing and development of talents possessing various professional skills. *(d)* The innovation ecosystem pertains to an organic system established for the cultivation and development of skilled talents with technological innovation capabilities. *(e)* The service and support ecosystem denote an organic system providing comprehensive support and services to skilled talents. This includes various aspects such as employment and entrepreneurship training, job services, career development, government policies, aiming to offer skilled talents a favorable working environment, developmental opportunities, and welfare benefits.

"Skill Talent Ecology Evaluation Indicator System" (see Table 1 for details), consisting of 5 primary indicators (A1-A5), 14 secondary indicators (B1-B14), and 34 tertiary indicators (C1-C34).

It is important to emphasize that within the "Kinetic Ecosystem (A3)", two additional indicators, "Talent Transition Potential within the Field (C21)" and "Talent Transition Potential across Fields (C22)," require supplementary calculations. The coupling coordination model can multidimensionally assess the interactions and connections between different systems, and also explain the dynamic relationships among these systems [24]. It can reflect the dynamic changes of skilled talents across various industries. "Talent Transition Potential within the Field" refers to the degree of free coordination in talent transition and exchange between industry domains where skilled talents are relatively concentrated. Based on data availability, this study selected five major industries where skilled talents are concentrated, namely "Information Technology Services, Finance, Manufacturing, Culture, Sports and Entertainment, and Construction." The coupling coordination was calculated pairwise between these five industries, and the average was taken to obtain the final "Intra-domain Talent Transition Potential."

Similarly, "Talent Transition Potential across Fields" refers to the degree of free coordination in talent transition and exchange between industries upstream and downstream of the industry where skilled talents are engaged. Again, based on data availability, this study selected four major industries, namely "Commodity Services, Wholesale and Retail, Leasing and Business Services, Transportation, Warehousing, and Postal Services." The calculation involved pairing each of the five major industries with these four upstream and downstream industries, calculating the coupling coordination, and taking the average. Due to the extensive calculations and space constraints, a detailed explanation is not provided here. The same approach applies to the measurements of "Intra-domain Coupling Coordination (C25)" and "Inter-domain Coupling Coordination (C26)."

This paper utilizes the measurement method of the "Coupling Coordination Model" to analyze the degree of coordination in development with other industries. The Coupling Coordination Model is widely applied in studying the coupling coordination relationships between multiple systems. The calculation formula is as follows:

$$D = (C*T)^{1/2}$$

$$C = 2*\{[(U_1*U_2)/(U_1 + U_2)^2]^{1/2}\}$$

$$T = A*U_1 + B*U_2$$

**Table 1. Theoretical model for skilled talent ecosystem evaluation (Including weighted results).**

| Primary Indicators | $\omega_A$ | Secondary Indicators | Single Ranking Weight | Tertiary Indicators | Synthesized Weight | Top10 |
|---|---|---|---|---|---|---|
| Cultivation Ecosystem A1 | 0.425 | Technical and Vocational Schools B1 | 0.404 | Number of Technical and Vocational Schools (C1) | 0.0368 | ① |
| | | | | Number of Faculty Members (C2) | 0.0319 | ⑥ |
| | | | | Number of Employed Students (C3) | 0.0291 | |
| | | | | Number of Obtained Vocational Qualifications (C4) | 0.0293 | |
| | | Employment Training B2 | 0.314 | Number of Skill Training Institutions (C5) | 0.0322 | ④ |
| | | | | Number of Faculty Members in Training Institutions (C6) | 0.0295 | |
| | | | | Number of Participants in Skill Training (C7) | 0.0305 | ⑩ |
| | | | | Number of Obtained Vocational Qualifications in Training Institutions (C8) | 0.0296 | |
| | | Private Vocational Skills Training B3 | 0.282 | Number of Skill Training Institutions (C9) | 0.0342 | ② |
| | | | | Number of Faculty Members in Training Institutions (C10) | 0.0316 | ⑦ |
| | | | | Number of Participants in Skill Training (C11) | 0.0320 | ⑤ |
| | | | | Number of Obtained Vocational Qualifications in Training Institutions (C12) | 0.0291 | |
| Potential Ecosystem A2 | 0.173 | Skilled Talent Stock B4 | 0.222 | Talent Stock (C13) | 0.0296 | |
| | | | | Percentage of Talent Stock (%) (C14) | 0.0274 | |
| | | Economic Contribution B5 | 0.486 | GDP (in thousands of yuan) (C15) | 0.0312 | ⑧ |
| | | | | Percentage of Total Output Value (%) (C16)(Percentage of Output Value in Manufacturing) | 0.0304 | |
| | | Capability Value B6 | 0.292 | Average Salary Income (in yuan) (C17) | 0.0291 | |
| | | | | Average Vocational Skill Level (C18) | 0.0302 | |
| Kinetic Ecosystem A3 | 0.147 | Growth Rate of Skilled Talent Stock B7 | 0.212 | Increment of Talent (C19) | 0.0281 | |
| | | | | Growth Rate of Talent (%) (C20) | 0.0285 | |
| | | Talent Transition Potential B8 | 0.204 | Talent Transition Potential within the Field (C21) | 0.0121 | |
| | | | | Talent Transition Potential across Fields (C22) | 0.0086 | |
| | | Output Growth Rate B9 | 0.352 | Increment of Output Value (in thousands of yuan) (C23) | 0.0327 | ③ |
| | | | | Growth Rate of Output Value (%) (C24) | 0.0306 | ⑨ |
| | | Industrial Activity B10 | 0.235 | Coordination Degree within the Field (C25) | 0.0265 | |
| | | | | Coordination Degree across Fields (C26) | 0.0242 | |
| Service and Support Ecosystem A4 | 0.162 | Incubation Centers B11 | 0.417 | Number of Innovation and Entrepreneurship Bases, Co-working Spaces (C27) | 0.0298 | |
| | | | | Number of Service-oriented Skilled Talents (C28) | 0.0297 | |
| | | Policy Support B12 | 0.583 | Number of Incentive Policies for Skilled Talents (C29) | 0.0283 | |
| | | | | Financial Expenditure on Science and Technology (in thousands of yuan) (C30) | 0.0265 | |
| Innovation Ecosystem A5 | 0.093 | Research and Development Investment B13 | 0.362 | R&D Investment Amount (in thousands of yuan) (C31) | 0.0261 | |
| | | | | R&D as a Percentage of GDP (%) (C32) | 0.0268 | |
| | | Skill Patents B14 | 0.221 | Number of Granted Skill Patents (C33) | 0.0255 | |
| | | | | Transaction Volume in the Technology Market (in thousands of yuan) (C34) | 0.0246 | |

Data Sources: Statistical Yearbooks and Economic Yearbooks of various provinces in China (2010-2022); Provincial Statistical Bulletins on National Economic and Social Development (2010-2022); Official websites of provincial statistical bureaus; Official website of the National Bureau of Statistics; Official websites of provincial departments of human resources and social security; "China Labor Statistics Yearbook (2010-2022)" and "China Population and Employment Statistics Yearbook (2010-2022)" from provincial labor departments

In the equations, where $D$ represents the Coupling Coordination Degree, with a range of [0, 1]. Alarger $D$ indicates a more coordinated development level between the two industries, with bettervitality. A smaller $D$ suggests an incongruent development level and lower vitality between the two industries. $C$ stands for Coupling Degree, with a range of [0, 1]. A higher $C$ implies a bettercoupling status between the two industries, while a smaller $C$ indicates a poorer coupling status,tending towards disordered development. $T$ represents the comprehensive coordination indexbetween the two industries, and $A$ and $B$ are the respective proportions of employees in the twoindustries (calculated here as the ratio of GDP between the two industries, similarly for thenumber of employed individuals).

## 3.2 "AHP-entropy method" evaluation model

This paper employs the AHP-Entropy Method, which is designed to overcome the issues of subjective assessment and weight determination in traditional AHP methods. To address this limitation, the entropy method is introduced, determining weights based on the information entropy of the data itself, thereby reducing subjective interference and enhancing the objectivity and scientific validity of the weights.

## 3.3 Neural network model

Accurately grading the skill talent ecosystem is a complex problem. There are numerous factors influencing the skill talent ecosystem, and they intertwine, permeate, and mutually influence each other, making it challenging to describe using a deterministic mathematical model. To grade the various indicators of the skill talent ecosystem, this paper constructs a Discrete Hopfield Neural Network (DHNN) model:

In the Discrete Hopfield Network, binary neurons are employed, and thus, the output discrete values of 1 and -1 respectively represent neurons in an activated and inhibited state [25]. DHNN is a single-layer, binary output feedback network. The structure of the Discrete Hopfield Neural Network composed of three neurons is illustrated in the figure below. In this figure, the 0th layer serves as the input to the network, it is not an actual neuron, and therefore, it has no computational function. The 1st layer consists of neurons, executing the sum of products of input information and weight coefficients, and generating output information after being processed by a nonlinear function, denoted as f. The function f is a simple threshold function: if the output information of the neuron exceeds the threshold $\theta$, then the output value of the neuron is 1; if it is less than the threshold $\theta$, then the output value is -1.

For binary neurons, the computation formula is as follows:

$$u_j = \sum_i w_{ij} y_i + x_i$$

The formula is as follows, where $x_j$ represents the external input. Additionally, there is:

$$\begin{cases} y_i = 1, & u_j \geq \theta_j \\ y_i = -1, & u_j < \theta_j \end{cases}$$

The state of a DHNN is a collection of output neuron information. For a network with an output layer consisting of $n$ neurons, its state at time $t$ is an $n$-dimensional vector:

$$Y(t) = [y_1(t), y_2(t), y_3(t), \ldots, y_n(t)]$$

Since $y_i(t)$ $(i=1, 2,\ldots,n)$ can take values of 1 or -1, the $n$-dimensional vector $Y(t)$ has $2^n$ states,

meaning the network has $2^n$ states. Considering the general node state of DHNN, where $y_j(t)$ represents the state of the j-th neuron, i.e., node j at time t, the state of the node at the next time step *(t+1)* can be obtained as follows:

$$y_j(t+1) = f[u_j(t)] = \begin{cases} 1, & u_j(t) \geq 0 \\ -1, & u_j(t) < 0 \end{cases}$$

If $w_{ij}$ is equal to 0 when *i=j*, it indicates that the output of a neuron does not feed back into its own input. In this case, the DHNN is referred to as a network without self-feedback.

$$u_j(t) = \sum_{i=1}^{n} w_{ij} y_i(t) + x_j - \theta_j$$

If $w_{ij}$ is not equal to 0 when *i=j*, it indicates that the output of a neuron will be fed back into its input. In this case, the DHNN is referred to as a network with self-feedback [26]. This study aims to classify various levels of talent ecosystem indicators and model the corresponding evaluation indicators as equilibrium points of a Discrete Hopfield Neural Network.

### 3.4 PVAR model

Given that the impact of digital transformation on the skill talent ecosystem has a certain degree of lag, and to facilitate the observation of the trend of the impact, this paper selects the Panel Vector Autoregressive model (PVAR) for data estimation and analysis [27]. The model formulation is as follows:

$$Y_{it} = \theta_0 + \sum_{j=1}^{n} \theta_j Y_{i,t-j} + \mu_i + \nu_t + \varepsilon_{it}$$

The model consists of a 1x3 column vector, encompassing four endogenous variables: "Cultivation Ecology," "Potential Ecology," "Kinetic Ecology," "Innovation Ecology," and "Service and Support Ecology." Here, $\theta_0$ represents the intercept term, *j* denotes the lag order (sufficiently capturing the lagged effects of external environmental factors), $\theta_j$ is the parameter matrix for the *j*-th lag order. $\mu_i$ represents individual fixed effects, $\nu_t$ stands for individual time effects, and $\varepsilon_{i,t}$ denotes the random disturbance term (see Table 2 for details).

Considering the availability of data, this section will conduct heterogeneity analysis focusing on five major industries: information technology services, finance, manufacturing, cultural, sports, and entertainment industries, and construction industry.

## 4 Analysis of model results

### 4.1 Results of the evaluation weights for skill talent ecology

According to the calculation steps of the Analytic Hierarchy Process (AHP), a confidence coefficient of 0.6 (moderate fuzziness) and an optimism coefficient of 0.6 (neutral risk preference) were chosen. Combining expert judgment data, the final judgment matrix was obtained. The maximum eigenvalue ($A_{max}$) was calculated as 3.168, and the corresponding eigenvector was [0.425, 0.173, 0.147, 0.162, 0.093]. The consistency test results showed that all pairwise matrices passed.

Based on this algorithm, the skill talent ecology evaluation indicator weight results were obtained, as shown in **Table 1**. Among the five primary indicators of skill talent ecology, the weight of Cultivation Ecology (A1) was the highest (0.425), followed by Potential Ecology (A2) with a weight of 0.173. Support and Service Ecology (A4) ranked third with a weight of 0.162,

**Table 2. Model variables and data sources.**

| Category | Variable | Measurement Indicator | Data Source |
|---|---|---|---|
| Dependent Variable | Cultivation Ecology | See indicator system construction for details (Table 1) | Data sources include the statistical yearbooks of each province from 2013 to 2020, the national economic and social development statistical bulletins of each province, and official data from the National Bureau of Statistics, among others. |
| | Potential Ecology | | |
| | Kinetic Ecology | | |
| | Innovation Ecology | | |
| | Service and Support Ecology | | |
| Core Explanatory Variable | Digital Transformation | "Digital Transformation" is synthesized from three indicators: "Information Technology Development," "Internet Development," and "Digital Transaction Development." | |
| Control Variable | Total Fixed Asset Investment | Total Fixed Asset Investment of the Whole Society (in RMB) | |
| | Value Added of the Primary Industry | Added Value of the Primary Industry (in RMB) | |
| | Foreign Enterprise Investment | Total Investment of Foreign-Invested Enterprises (in RMB) | |
| | Information Technology Service Industry | Added Value of the Manufacturing Industry (in RMB) | |
| | Financial Industry | Added Value of the Financial Industry (in RMB) | |
| | Manufacturing Industry | Added Value of the Manufacturing Industry (in RMB) | |
| | Culture, Sports, and Entertainment Industry | Added Value of the Culture, Sports, and Entertainment Industry (in RMB) | |
| | Construction Industry | Added Value of the Construction Industry (in RMB) | |

followed by Kinetic Ecology (A3) with a weight of 0.147, and finally, Innovation Ecology (A5) had the smallest weight of 0.093. The ranking of primary indicators is as follows: Cultivation Ecology A1 > Potential Ecology A2 > Support and Service Ecology A4 > Kinetic Ecology A3 > Innovation Ecology A5.

## 4.2 Hopfield neural network model – level evaluation

In the academic community, there is no definitive standard for the evaluation of the skill talent ecology. Therefore, it is challenging to establish precise numerical values as grading criteria. This study categorizes it into five levels, creating a set of evaluation standards V, V={v1, v2, v3, v4, v5}={A (Excellent), B+ (Good), B- (Fair), C+ (Poor), C (Very Poor)}. Furthermore, to facilitate the subsequent graded evaluation of talent ecology indicators, based on the scores provided by 11 experts, and according to the weight of each indicator, a grading standard is established.

Based on the weight results of the skill talent ecology evaluation indicators, a set of weights for the evaluation indicators is established: Z=(0.425, 0.173, 0.147, 0.162, 0.093); Y1=(0.404, 0.314, 0.282); Y2=(0.292, 0.486, 0.222); Y3=(0.312, 0.453, 0.235); Y4=(0.417, 0.583); Y5=(0.362, 0.221, 0.417).

Since the states of the discrete Hopfield neural network neurons are only 1 and -1, when mapping the evaluation indicators to the states of the neurons, encoding is required. The encoding rule is as follows: when the indicator value is greater than or equal to a certain level, the corresponding neuron state is set to "1"; otherwise, it is set to "-1". The ideal encoding of the five-level evaluation indicators is illustrated in Fig 2, where ○ represents the neuron state as "1," indicating that it is greater than or equal to the corresponding level's ideal evaluation indicator value, and ● represents the opposite.

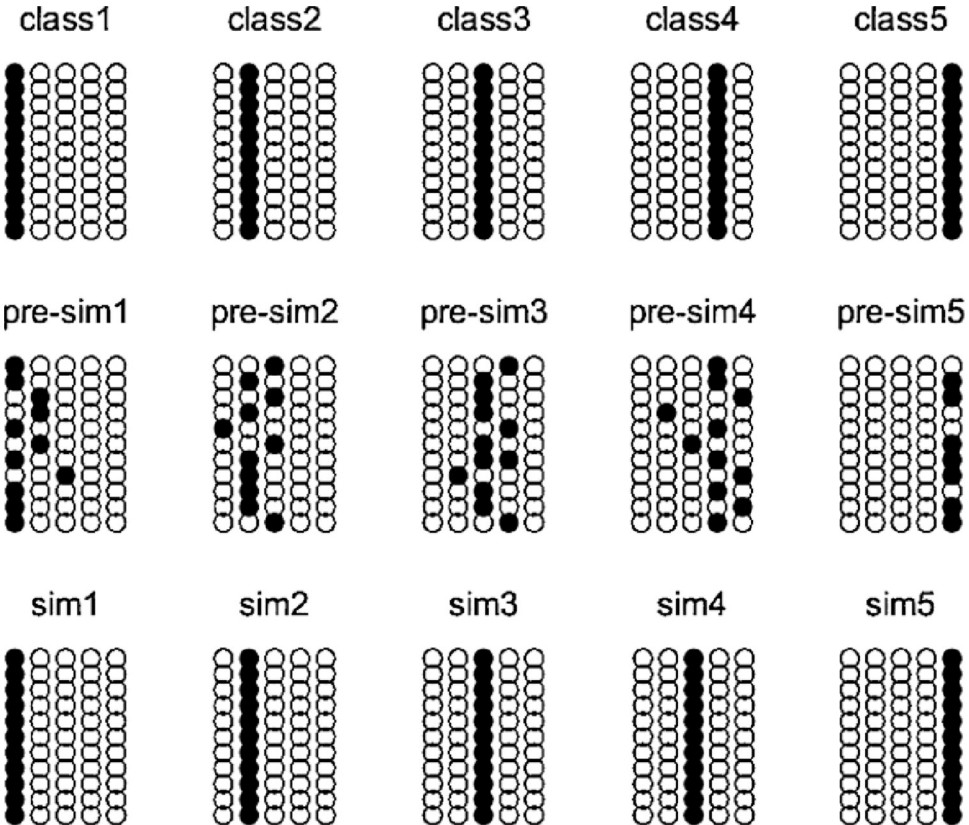

**Fig 2. The graded evaluation results of the talent ecology.**

This article utilizes functions provided by the MATLAB Neural Network Toolbox to implement the design steps gradually within the MATLAB environment. The simulation results are shown in Fig 2. In this figure, the first row represents the encoding of the five ideal talent ecology level evaluation indicators. The second row is the encoding of the talent ecology evaluation indicators to be graded. The third row shows the classification results of the Hopfield neural network analysis, where sim1 corresponds to the Cultivation Ecology (A1), sim2 to the Potential Energy Ecology (A2), sim3 to the Kinetic Energy Ecology (A3), sim4 to the Service and Support Ecology (A4), and sim5 to the Innovation Ecology (A5) (See **Fig 2** for details).

The designed Hopfield network is clearly effective in classification and evaluation, aligning well with the preceding weight analysis results. Consequently, it to some extent reflects the degree and current status of the talent ecology. Based on the results from the Hopfield neural network analysis, the graded evaluation of the talent ecology is as follows: Cultivation Ecology (A1) is rated as A (Excellent), Potential Ecology (A2) as B+ (Good), Kinetic Ecology (A3) as B- (Fair), Service and Support Ecology (A4) as B- (Fair), and Innovation Ecology (A5) as C (Poor).

## 4.3 PVAR model - Impact trend analysis

**4.3.1 Pulse response results of digital transformation on cultivation ecology.** The trend line of the impact of digital transformation on cultivation ecology is not within a reasonable confidence interval, and the impact trend of digital transformation on cultivation ecology is not significant (due to space constraints, it is not further presented). It might take more time

for digital transformation to truly change the ecology of skill talent cultivation. The impact of digital transformation may be more evident in the profound changes in the education system, such as updates in teaching methods, curriculum content, and improvements in educational tools.

**4.3.2 Pulse response results of digital transformation on potential ecology.** According to the results of the pulse impact of digital transformation on potential ecology in **Fig 3**., there is a significant positive feedback in the potential ecology of skill talent. This feedback reaches its maximum between periods 0 and 1, gradually decreasing afterward. This indicates that, on the one hand, digital transformation typically involves the introduction of new technologies and working methods, leading to an increased demand for new skills and knowledge and a decreased demand for existing skills. This, in turn, affects the stock of skill talent. At the same time, digital transformation results in a redefinition of the economic contribution of skill talent.

**4.3.3 Pulse response results of digital transformation on kinetic ecology.** According to the results of the pulse impact of digital transformation on kinetic ecology in **Fig 4**, there is a significant positive feedback in the kinetic ecology of skill talent. This feedback reaches its maximum between periods 0 and 1, gradually decreasing afterward. This indicates that digital transformation typically requires more talents with skills in digital technology, data science, and innovation, leading to a rapid growth in the demand for skill talent. Simultaneously, it may also impact traditional industries, potentially causing changes in industrial activity, improving production efficiency, and increasing output value.

**4.3.4 Pulse response results of digital transformation on innovative ecology.** According to the pulse impact results of digital transformation on innovative ecology in **Fig 5**, there is a significant positive feedback in the innovative ecology of skill talent. This feedback reaches its maximum between periods 0 and 1, gradually decreasing afterward. This indicates that digital transformation brings about rapid technological advancements, requiring skill talents to continue learning and developing new skills to stay competitive and adapt to new digital tools and platforms.

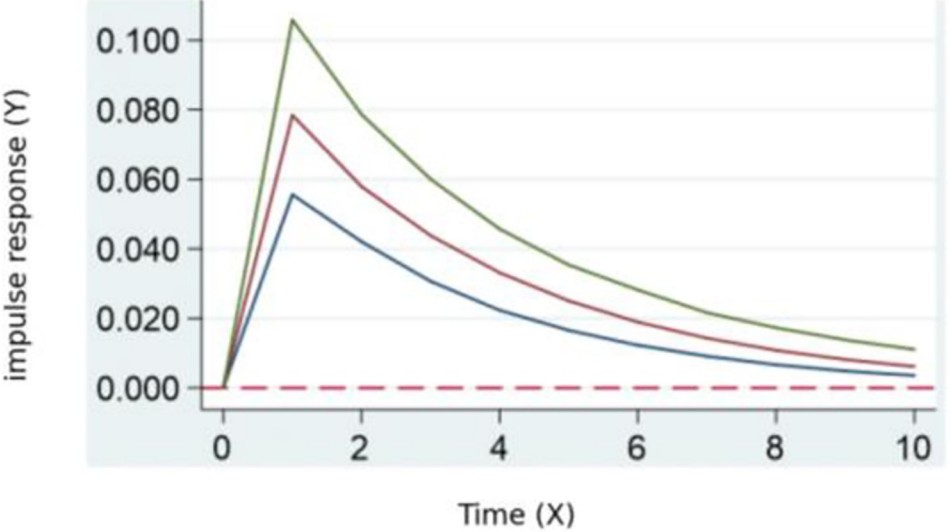

**Fig 3. Digital transformation's pulse impact on potential ecology.**

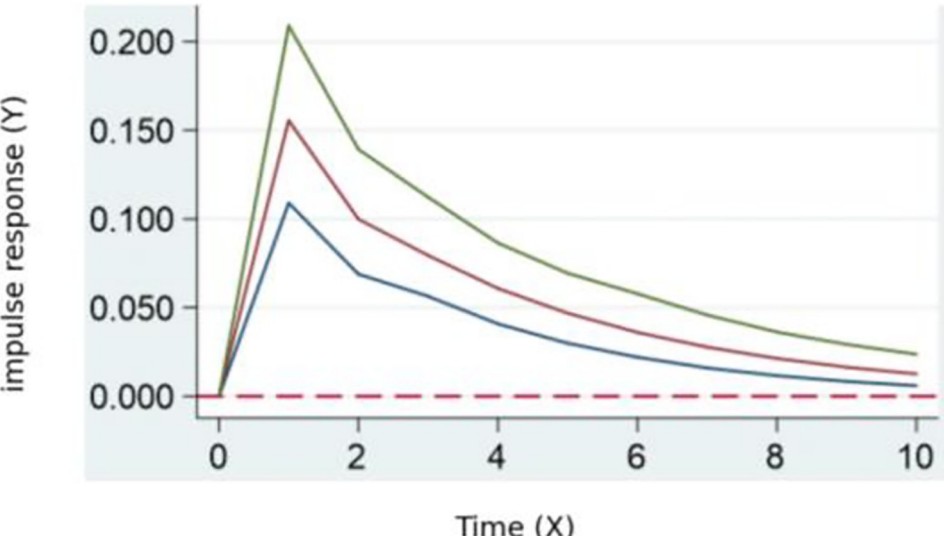

**Fig 4. Digital transformation's pulse impact on kinetic ecology.**

### 4.3.5 Pulse response results of digital transformation on service and support ecology.

According to the pulse impact results of digital transformation on the service and support ecology in **Fig 6**, there is a significant positive feedback in the service and support ecology of skill talent. This feedback reaches its maximum between periods 0 and 1, gradually decreasing afterward. This indicates that the stimulus of digital transformation on the service and support ecology will promote the increase of digital entrepreneurship. Incubation spaces and industrial parks provide resources and infrastructure for these startups to conduct innovative work, leading to a potential increase in demand for such locations. This also includes policy incentives, benefits, and funding to support skill talents working in the digital field.

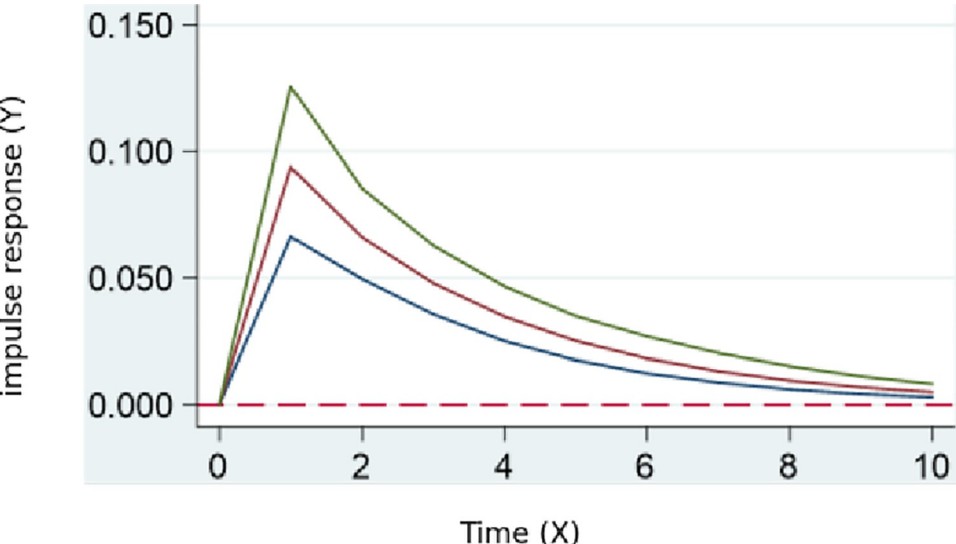

**Fig 5. Digital transformation's pulse impact on innovative ecology.**

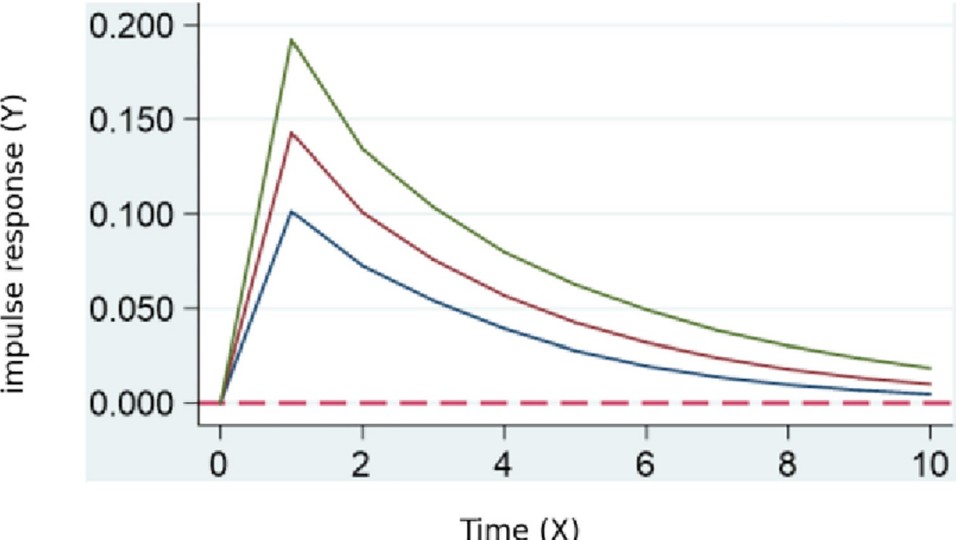

**Fig 6. Digital transformation's pulse impact on service and support ecology.**

## 5 Conclusion and outlook

### 5.1 Conclusion

Conclusion 1: Based on the "AHP-Entropy Method," the conclusion is that in the skill talent ecology, the cultivation ecology is relatively good, vocational schools have advantages, and talent reserves are sufficient. Economic contribution shows superiority, but talent stock still needs improvement. Output growth rate is significant, but industrial activity needs strengthening, and there is potential for cross-industry collaboration. Policy support is robust, and the digitalization of the innovation ecology is evident, but the development of skill patents needs enhancement. Some scholars also believe that the talent ecosystem changes in sync with technological progress and economic growth, and that a positive talent ecosystem has a positive feedback effect [28]. The comprehensive ecological development of skill talents needs to be further promoted, especially in aspects such as talent stock, industrial activity, and technological innovation.

Conclusion 2: According to the level evaluation results analyzed by the Hopfield neural network, the cultivation ecology is rated as A (Excellent), the potential ecology is rated as B+ (Good), the kinetic ecology is rated as B- (Fair), the service and support ecology is rated as B- (Fair), and the innovation ecology (A5) is rated as C (Poor). Similarly, some scholars have found that innovation capability or an innovative environment significantly enhances the talent ecosystem [29]. This indicates that innovation remains the core area of talent development.

Conclusion 3: Based on the regression results of the PVAR model, the potential ecology, kinetic ecology, innovation ecology, and service and support ecology all show significant positive feedback. The maximum effect of digital transformation occurs between periods 0 and 1 and gradually decreases. This indicates that digital transformation drives an increase in demand for new skills, impacting talent stock and economic contribution methods. In the kinetic ecology, the increased demand for digital transformation may impact traditional industries, raising output. In the innovation ecology, skill talents need to continuously learn new skills to adapt to rapidly evolving technologies. In the service and support ecology, digital entrepreneurship increases, requiring policy incentives and support from incubation centers.

### 5.2 Recommendations

(a) Enhancing the cultivation ecology of skilled talents comprehensively involves devising an integrated talent development plan, promoting industry-academic-research collaboration, providing interdisciplinary education and training, and fostering international exchange. (b) Consolidating the potential energy ecology entails continuous professional training, refining industry standards and certification systems, offering diverse career advancement mechanisms, and implementing performance-based compensation. (c) Activating the kinetic energy ecology involves establishing effective learning and development mechanisms, creating promotion channels, introducing feedback and evaluation systems, and elevating the social status of skilled talents. (d) Exploring the innovation ecology includes establishing innovation labs and practice bases, fostering craftsmanship spirit, instituting innovation rewards and support, and enhancing digital literacy. (e) Upgrading the service and support ecology encompasses establishing an intelligent career counseling system, building interdisciplinary talent exchange platforms, implementing comprehensive training plans, and establishing a lifelong development system for skilled talents.

Shaping a comprehensive digital-era skill talent ecology ensures diverse knowledge, innovation, continuous learning motivation, and societal recognition, contributing to meeting evolving digital demands and promoting sustainability.

### 5.3 Outlook

The future of skill talent ecology will exhibit diverse characteristics, and evaluation methods need continuous improvement to adapt to changes. Firstly, the application of big data and artificial intelligence technologies represents a forward-looking evaluation method applicable across industries to address data challenges and enhance accuracy. Secondly, personalized evaluation and training programs are crucial, catering to the specific needs of different industries to boost competitiveness and inspire learning motivation. Thirdly, strengthening cross-disciplinary collaboration is a future development direction, integrating expertise from various fields to promote the refinement of evaluation indicators and drive skill talent development across disciplines. Continuous improvement of evaluation methods is essential in the future, utilizing big data, personalized evaluation, and cross-disciplinary collaboration to enhance competitiveness, propel the development of skill talent ecology, and contribute to societal progress. Governments, industries, academia, and various sectors of society should collaborate to provide support and investment for the future development of skill talent ecology.

## Supporting information

**S1 Data.**
(XLS)

## Author Contributions

**Conceptualization:** Minqiang Xing.

**Data curation:** Gaoyang Liang.

**Methodology:** Gaoyang Liang.

**Software:** Gaoyang Liang.

**Writing – original draft:** Minqiang Xing.

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
