## [Decision Letter · Decision Letter 0]

21 Feb 2024

PONE-D-23-38696Research on the Evaluation and Impact Trends of China's Skill Talent Ecosystem in the Digital Era – An Analysis Based on Neural Network Models and PVAR ModelsPLOS ONE

Dear Dr. gaoyang, Thank you for submitting your manuscript to PLOS ONE. After careful consideration, we feel that it has merit but does not fully meet PLOS ONE’s publication criteria as it currently stands. Therefore, we invite you to submit a revised version of the manuscript that addresses the points raised during the review process.

**  **

**Please add more related literature**

**update your reference**

**compare with other works****language edit****make more related useful figures** Please submit your revised manuscript by Apr 06 2024 11:59PM. If you will need more time than this to complete your revisions, please reply to this message or contact the journal office at plosone@plos.org. Please include the following items when submitting your revised manuscript:A rebuttal letter that responds to each point raised by the academic editor and reviewer(s). You should upload this letter as a separate file labeled 'Response to Reviewers'.A marked-up copy of your manuscript that highlights changes made to the original version. You should upload this as a separate file labeled 'Revised Manuscript with Track Changes'.An unmarked version of your revised paper without tracked changes. You should upload this as a separate file labeled 'Manuscript'.

We look forward to receiving your revised manuscript.

Kind regards,

Amirsalar Khandan, Ph.D.

Academic Editor

PLOS ONE

Journal Requirements:

This paper was supported by the National Natural Science Foundation of China [No. 71872142].

Reviewers' comments:

Reviewer's Responses to Questions

**Comments to the Author**

1. Is the manuscript technically sound, and do the data support the conclusions?

Reviewer #1: Yes

2. Has the statistical analysis been performed appropriately and rigorously? 

Reviewer #1: Yes

3. Have the authors made all data underlying the findings in their manuscript fully available?

Reviewer #1: Yes

4. Is the manuscript presented in an intelligible fashion and written in standard English?

Reviewer #1: Yes

5. Review Comments to the Author

Reviewer #1: 1. Mention the significance of coupling coordination model.

2. Figures 3 and 4 are incomplete. Mark X and Y axis properly.

3. Add few recent references related to Neural Network and PVAR models.

4. Manuscript is checked for typographic and grammatical errors.

6. PLOS authors have the option to publish the peer review history of their article (what does this mean?). If published, this will include your full peer review and any attached files.

Reviewer #1: No

---

## [Author Response · Author response to Decision Letter 0]

12 Mar 2024

Response to Reviewers：

Q1: Mention the significance of coupling coordination model.

Supplemented the importance and rationale for using the coupling coordination model.

Q2: Figures 3 and 4 are incomplete. Mark X and Y axis properly.

Figures 3 and 4 have been corrected, and the X and Y axes have been added.

Q3: Add few recent references related to Neural Network and PVAR models.

Relevant literature on the PVAR model and neural network model has been added.

Q4: Compare with other works

In the conclusion analysis section, compare and analyze the research conclusions of other scholars.

Summary:

Add more relevant literature.

Update the reference list.

Compare and analyze results with those of related papers.

Make language adjustments and improvements.

Correct any imperfections in figures and tables.

Thank you for your expert advice!

---

## [Decision Letter · Decision Letter 1]

16 Apr 2024

Research on the Evaluation and Impact Trends of China's Skill Talent Ecosystem in the Digital Era – An Analysis Based on Neural Network Models and PVAR Models

PONE-D-23-38696R1

Dear Dr.  Gaoyang,

We’re pleased to inform you that your manuscript has been judged scientifically suitable for publication and will be formally accepted for publication once it meets all outstanding technical requirements.

Kind regards,

Amirsalar Khandan, Ph.D.

Academic Editor

PLOS ONE

Additional Editor Comments (optional):

Reviewers' comments:

Reviewer's Responses to Questions

**Comments to the Author**

1. If the authors have adequately addressed your comments raised in a previous round of review and you feel that this manuscript is now acceptable for publication, you may indicate that here to bypass the “Comments to the Author” section, enter your conflict of interest statement in the “Confidential to Editor” section, and submit your "Accept" recommendation.

Reviewer #1: All comments have been addressed

2. Is the manuscript technically sound, and do the data support the conclusions?

Reviewer #1: Yes

3. Has the statistical analysis been performed appropriately and rigorously? 

Reviewer #1: Yes

4. Have the authors made all data underlying the findings in their manuscript fully available?

Reviewer #1: Yes

5. Is the manuscript presented in an intelligible fashion and written in standard English?

Reviewer #1: Yes

6. Review Comments to the Author

Reviewer #1: All the comments have been properly addressed. Manuscript can be accepted for publication in PLOS ONE

7. PLOS authors have the option to publish the peer review history of their article (what does this mean?). If published, this will include your full peer review and any attached files.

Reviewer #1: No

---

## [Editor Report · Acceptance letter]

1 May 2024

PONE-D-23-38696R1 

PLOS ONE

Dear Dr. Liang, 

I'm pleased to inform you that your manuscript has been deemed suitable for publication in PLOS ONE. Congratulations! Your manuscript is now being handed over to our production team.

Kind regards, 

on behalf of

Dr. Amirsalar Khandan 

Academic Editor

PLOS ONE